# Thermo-optic epsilon-near-zero effects

Jiaye Wu [1] ✉, Marco Clementi [1], Chenxingyu Huang[2,3], Feng Ye [2], Hongyan Fu [3], Lei Lu [2], Shengdong Zhang [2], Qian Li [2] ✉ & Camille-Sophie Brès [1] ✉

Nonlinear epsilon-near-zero (ENZ) nanodevices featuring vanishing permittivity and CMOS-compatibility are attractive solutions for large-scale-integrated systems-on-chips. Such confined systems with unavoidable heat generation impose critical challenges for semiconductor-based ENZ performances. While their optical properties are temperature-sensitive, there is no systematic analysis on such crucial dependence. Here, we experimentally report the linear and nonlinear thermo-optic ENZ effects in indium tin oxide. We characterize its temperature-dependent optical properties with ENZ frequencies covering the telecommunication O-band, C-band, and 2-$\mu$m-band. Depending on the ENZ frequency, it exhibits an unprecedented 70–93-THz-broadband 660–955% enhancement over the conventional thermo-optic effect. The ENZ-induced fast-varying large group velocity dispersion up to 0.03–0.18 fs²nm⁻¹ and its temperature dependence are also observed for the first time. Remarkably, the thermo-optic nonlinearity demonstrates a 1113–2866% enhancement, on par with its reported ENZ-enhanced Kerr nonlinearity. Our work provides references for packaged ENZ-enabled photonic integrated circuit designs, as well as a new platform for nonlinear photonic applications and emulations.

The recent decade has seen the prosperous advances of epsilon-near-zero (ENZ) photonics[1–3]. Due to a vanishing permittivity at the vicinity of the ENZ frequency, these exotic materials can exhibit linear properties like slow light[4], phase tunneling[5], and electric field enhancement[6]; as well as nonlinear ones like nonlinearity enhancement[7–9], pulse shaping[10,11], and frequency translation[12,13]. The unconventional optical behaviors of ENZ materials are also found to be analogous to other physical systems, such as electro-magnetic ideal fluids[14,15], temporal refraction[13], radio-frequency (RF) superconducting quantum interference device[16], and temporal double-slit phenomena[11,17], allowing further experimental explorations to be migrated and performed at spectral ranges that are previously hard to reach or even impossible on their original platforms (e.g., the electrical RF frequencies of megahertz to gigahertz versus the optical frequencies of hundreds of terahertz) and potentially a deeper understanding on the emulated physical phenomena.

Among the various mechanisms to realize ENZ behavior, a widely used group of plasmonic materials called transparent conducting oxides (TCOs) offers an effective, relatively low-cost, and versatile solution. In the TCO family, indium tin oxide (ITO), which can be ubiquitously found in modern microelectronic devices, is reported to show unprecedentedly high optical nonlinearities[7] and compatibility with the state-of-the-art complementary metal–oxide–semiconductor (CMOS) technology[18]. These traits allow ITO to perform as a free-space optical plug-in to realize terahertz emission[19,20], harmonic generation[21–24], and intra-cavity mode-lock wavelength re-selection[16], as well as on-chip devices like modulators[18,25–27] and all-optical switches[28–30]. By far, most of the ENZ experiments are free-space or on a temperature-stabilized single-device chip exposed in the open air[3], the effect of temperature on linear and nonlinear optical properties is often neglected. However, this cannot be the case when moving to more complex systems such as in laser cavities[16,31] and in photonic integrated chips (PICs)[18].

[1]École Polytechnique Fédérale de Lausanne (EPFL), Photonic Systems Laboratory (PHOSL), STI-IEM, Station 11, Lausanne CH-1015, Switzerland. [2]School of Electronic and Computer Engineering, Peking University, Shenzhen 518055, China. [3]Tsinghua Shenzhen International Graduate School, Tsinghua University, Shenzhen 518055, China. ✉e-mail: jiaye.wu@epfl.ch; liqian@pkusz.edu.cn; camille.bres@epfl.ch

Due to their inherent compatibility (especially for TCOs) with CMOS technology and PIC designs, ENZ materials could find a place of choice in complex photonic system-on-chips (SoCs), such as on-chip light sources[32], cross-talk-prohibiting waveguide claddings[33], and data processing devices[16,18,28]. As they progress towards large-scale integration (LSI) and eventually end-products, the inevitable heat generation by electrical auxiliary components and photonic micro-nano circuits could have a prominent impact on the optical properties and overall performance of integrated ENZ devices. As a relevant reference, modern microelectronic chips operate at a temperature range higher than room temperature and lower than the maximum junction temperature ($T_{junction(max)}$), which is usually no hotter than 95–105 °C, to avoid failure[34]. This poses a potential challenge to ENZ TCO materials whose linear and nonlinear optical properties are known to be temperature-sensitive[35–38]. However, to the best of our knowledge, there is still no study or systematic data on the impact and engineering implication of such thermal effects on TCO degenerate semiconductors and ENZ integrated optics.

Here, we experimentally report a series of thermo-optic ENZ effects by fine thermal engineering for the first time. We demonstrate the characteristics of temperature-dependent linear and nonlinear-related optical properties induced by thermo-optic ENZ effects below $T_{junction(max)}$, covering the major ENZ frequencies from the telecommunication O-band, C-band, and the 2-μm-band using ENZ ITO nanolayers. We observe a significant (660–955%) enhancement over the conventional thermo-optic effect in the corresponding ENZ regimes, with an ultrabroad bandwidth of 70–93 THz. Near the ENZ frequencies, the temperature-dependent fast-varying large group velocity dispersion (GVD) up to 0.03–0.18 fs$^2$ nm$^{-1}$ is also experimentally verified for the first time, values in agreement with previously reported theoretical works and several orders of magnitude larger than conventional optical materials. We finally show that the thermo-optic nonlinearity, resembling its Kerr nonlinearity counterpart, is also significantly enhanced in the ENZ regime over the conventional non-ENZ ITO. Due to its large value and ultrafast pulse-width threshold, the effect of the thermo-optic nonlinearity can be dominant for ENZ-enabled integrated devices and PICs operating at sub-picosecond range. Besides providing critical information on linear and nonlinear ENZ thermo-optic effects, our work offers practical references for both optical (nanophotonic, nonlinear, and integrated) and thermal engineering design considerations, especially for the applications in packaged electro- and all-optical photonic LSI SoC products where heat generation from the electronic components and PICs is inevitable. Additionally, the revealed effects might provide a deeper understanding and insight in the field of nano- and nonlinear ENZ photonics, opening a pathway to novel optical applications of this emerging class of materials.

## Results

### Thermal response of the ENZ frequency: principles

The key signature of an ENZ material is its ENZ frequency, which is known to be temperature-dependent. This frequency is the reference point of its exotic linear and nonlinear optical properties and studying its thermal tuning behavior is instrumental in understanding ENZ enhancements on linear and nonlinear thermo-optic effects.

The ENZ frequency can be understood from Eq. (1). For TCOs like the ITO, the permittivity can be described by the classical Drude model[39]:

$$\varepsilon = \varepsilon_r + i\varepsilon_i = \varepsilon_b - \frac{\omega_p^2}{\omega^2 + \gamma^2} + i\frac{\omega_p^2\gamma}{(\omega^2 + \gamma^2)\omega}, \tag{1}$$

where the subscripts "r" and "i" denote the real and imaginary parts of the permittivity, $\varepsilon_b$ is the background high-frequency permittivity, $\omega$ is the angular frequency of light, $\omega_p = (Ne^2/\varepsilon_0 m^*)^{1/2}$ is the plasma frequency, and $\gamma$ is the damping rate related to the carrier mobility

$\mu$: $\gamma = e/(\mu m^*)$ with $e$ and $m^*$ being the charge and the effective mass of the electron, respectively. The temperature dependence of $\mu$ can be obtained by Hall effect measurement (See Supplementary Fig. 6). In the expression of $\omega_p$, $N$ is the free carrier concentration measured in cm$^{-3}$, $\varepsilon_0$ is the vacuum permittivity constant. In Eq. (1), with sufficiently high $N$ it is possible to produce a zero real permittivity at the ENZ frequency $\nu_{ENZ} = \omega_{ENZ}/(2\pi)$.

Considering Eq. (1), it is intuitive to think about tuning the ENZ frequency by altering its electronic properties, noting that $\omega_p$ (and thus, $\omega_{ENZ}$) has a quadratic relation with $N$. Electrical field effect[18,25,40,41] can instantly alter the surface $N$ of film, while abundant carriers only accumulate within a few nanometers' depth[27]. In contrast, thermal annealing[36,37,42] is an effective approach to modulate the $N$ of the whole bulk, tuning over a wide spectral range (hundreds of nanometers). However, thermal annealing is normally irreversible, and thus is typically implemented during the film preparation rather than as a dynamic $N$-tuning method.[42] Finally, supercritical fluid treatments have also been suggested. Unlike annealing, they require significantly lower temperatures and shorter time; but similar to annealing, specialized machines are needed[38].

An alternative solution to consider, given the high sensitivity of ENZ ITO to $N$, is low-temperature tuning. The feasibility and properties of such tuning method, which could rely simply on common sensor-heater feedback-loop typically used in PICs[43], is of practical importance given engineering constraints of PICs, especially when the ENZ material is expected to work with other narrowband photonic components.

ITO is regarded as an $n$-type degenerate semiconductor as it is very heavily doped ($N \approx 10^{19}$–$10^{21}$ cm$^{-3}$). The optical and electrical behavior of ENZ ITO relies extensively on defects and trap states[38], which can significantly impact $N$ compared to the classical Fermi-Dirac estimation in non-degenerate semiconductors. To provide a description of the thermo-optic properties of ENZs, we recall here the grain-boundary barrier model[44] and present an illustration of the mechanism behind the temperature dependence of $N$ (and thus, $\nu_{ENZ}$) in Fig. 1. Figure 1a shows the relation among the valence band $E_V$, conduction band $E_C$, and Fermi energy $E_F$ at room temperature. It is well known that the two major sources of free carriers in ITO are tin (Sn) activation at the indium (In) site and the oxygen vacancy[45]. Due to the mismatch of valence electron number between Sn$^{4+}$ and In$^{3+}$, the defects in the ITO lattice create trap states and provide free electrons. ITO possesses a polycrystalline structure and the energy spikes in Fig. 1 denote the grain boundary with grain size $d_1$. Within the sub-annealing region (Fig. 1b), with an increasing temperature, the oxygen (O$_2$) from ambient air starts to accelerate its diffusion into the ITO lattice through the grain boundaries. The higher the temperature, the stronger the diffusion. The diffused O$_2$ occupies the traps and reduces their number ($N_t$)[38], which decreases the number of free electrons $N$ they can provide in the metal-oxygen system (Supplementary Note 1.1). A decrease in $N$ will reduce $\omega_p$ which further results in a significant decrease in $\omega_{ENZ}$ ($\nu_{ENZ}$) to compensate for the zero permittivity, as per Eq. (1). Note that this is the opposite to the case in degenerate semiconductors without considering trap states. Within this region, the described diffusion process is reversible (the desorption of oxygen), and with a decreasing temperature $N_t$ will recover.

At a much higher temperature ($T > T_{anneal}$, Fig. 1c) and without a deliberate supply of oxygen flow, the ITO goes through the annealing process. Oxygen located at dangling bonds instead of lattice bonds tends to escape the bulk structure[38] and create more oxygen vacancies, which lead to more free carriers in the conduction band. Excessive heat can also permanently change the lattice structure by the process of crystallization[45]. When the degree of crystallinity in ITO increases with temperature, the size of the grain grows ($d_2 > d_1$) and this is irreversible. In the annealing case, oxygen (lattice or dangling) with weak bonding tends to escape in the form of O$_2$ and more dopants are activated. This ensures a permanent increase in $N$ and a rise in $\nu_{ENZ}$, which agrees with the reported results[36,37,42].

## Temperature-dependent ENZ frequencies

To investigate the linear and nonlinear optical properties of the thermo-optic ENZ effects, we design proper thermal holders and implement controls (see Methods) to keep the samples in the states shown in Fig. 1b. We prepare 4 different types of ITO samples (see Supplementary Note 6 for sample characterizations) shown in Fig. 2a with ENZ frequencies located in the spectral region of interest for telecom and processing: the O-band ($v_{ENZ} \approx 235.64$ THz), C-band ($v_{ENZ} \approx 193.83$ THz), 2-µm-band ($v_{ENZ} \approx 157.03$ THz), and non-ENZ ITO used as reference. The ITO layers' nominal thicknesses are 130 nm and they are deposited onto a 1.1-mm thick 2 cm × 2 cm silica glass. As can be seen from Fig. 2a and Eq. (1), the lower the carrier concentration is, the darker it appears.

Considering the practical application scenario, we set the highest temperature limit to 100 °C, close to the typical equilibrium temperature of packaged PICs. In most consumer electronic chip products, this is defined as the maximum tolerated p–n junction temperature $T_{junction(max)}$. We collect ellipsometry data every 5 °C starting from the sub-ambient 10 °C (see Methods). Then the samples are cooled to 60, 40, and 20 °C to assess their recoverability. We extracted $v_{ENZ}$ of the three types of ENZ samples at all temperatures and plotted them in Fig. 2b. For all samples, $v_{ENZ}$ decreases with increasing temperature as predicted by the grain-boundary barrier model. The O-band, C-band, and 2-µm-band samples have temperature modulation rates of $-8.45 \pm 0.35$ GHz K$^{-1}$, $-7.44 \pm 0.93$ GHz K$^{-1}$, and $-6.45 \pm 0.65$ GHz K$^{-1}$, respectively, as estimated from linear fit of the recorded trends. The low $v_{ENZ}$ samples tend to have lower temperature modulation/tuning rates.

The O-band and C-band samples recover after cooling down. However, the 2-µm-band one experiences a sudden increase of $v_{ENZ}$ from >80 °C to 100 °C. When it is cooled, its $v_{ENZ}$ is even higher than before heating. This situation indicates an increased and non-recoverable $N$, and implies that annealing actually occurs from >80 °C. The lower $N$ the ENZ ITO sample has, the more it is prone to be annealed at a lower temperature. In order to obtain higher $N$, such as for O-band and C-band operation, their dopants are already activated via annealing during the fabrication process. For these already-activated samples, it is harder to be further annealed, i.e., a higher annealing temperature threshold is observed. We expect that the C-band sample will have an annealing temperature threshold between those of the O-band and 2-µm-band samples. The detailed complete ellipsometry data, $\varepsilon_r$ and $\varepsilon_i$, are presented in Supplementary

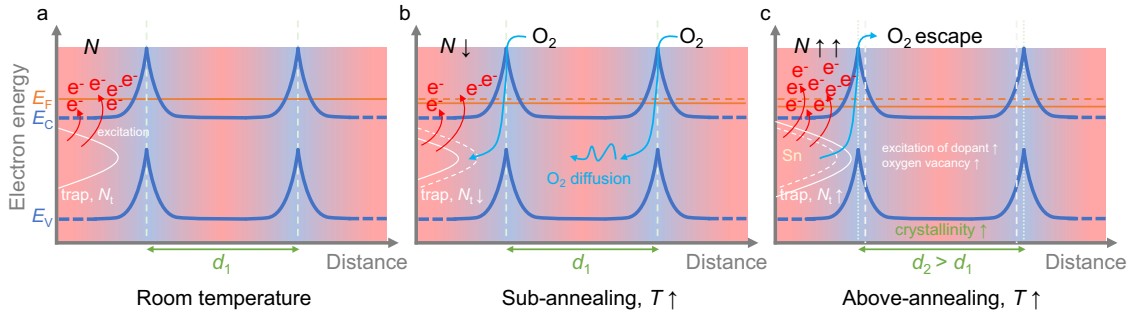

**Fig. 1 | Adapted grain-boundary barrier model in indium tin oxide at different temperature ranges. a** The energy levels of a degenerate semiconductor such as indium tin oxide. **b** Oxygen diffusion mechanism in the sub-annealing domain with increasing temperature. **c** Permanent structural change above the annealing threshold.

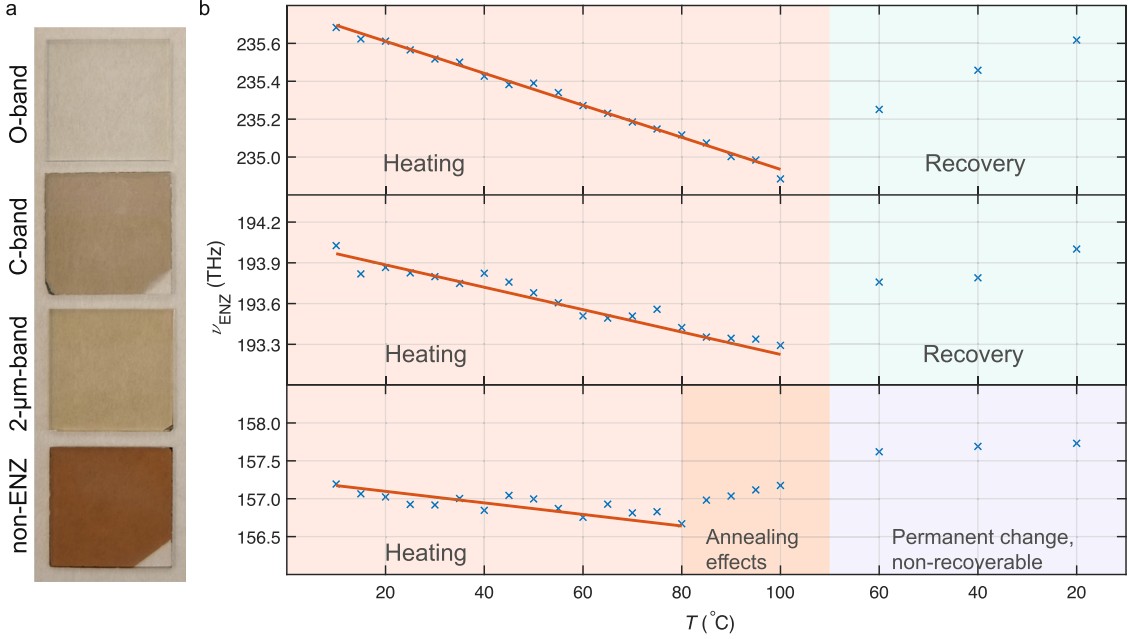

**Fig. 2 | Samples and their epsilon-near-zero (ENZ) frequency response. a** Indium tin oxide samples with ENZ frequency at O-band, C-band, 2-µm-band, and without ENZ effects. **b** The variation of ENZ frequency with temperature at different thermal phases of O-band (top), C-band (middle), and 2-µm-band (bottom) samples.

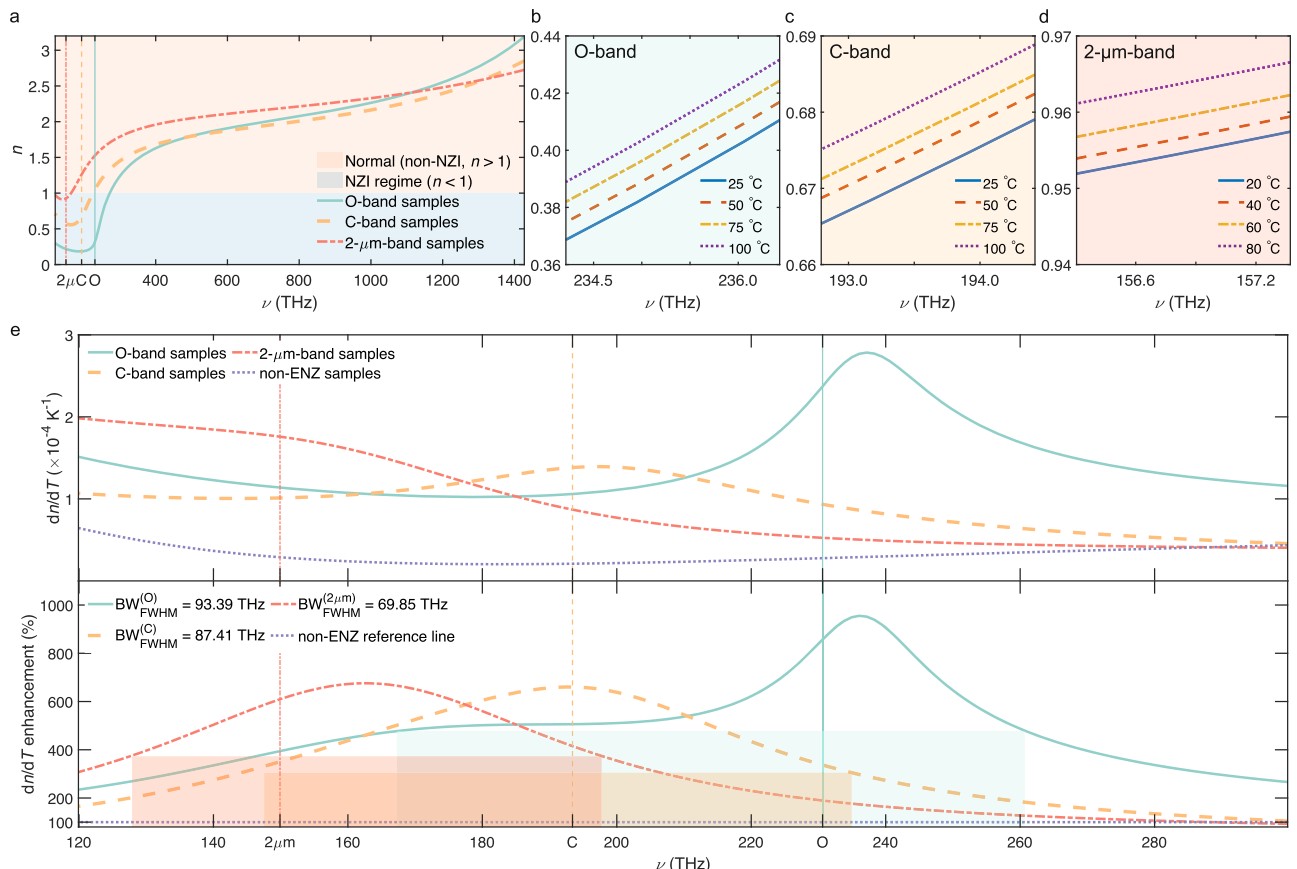

**Fig. 3 | Spectra of refractive index and the giant enhancement of thermo-optic effects in the epsilon-near-zero (ENZ) region. a** Full refractive index spectrum of the O-band, C-band, and 2-μm-band samples. IR stands for infrared, and UV is ultraviolet. NZI represents near-zero-index where $n < 1$. **b–d** The variations of refractive indices with temperature for these three sets of samples near the ENZ frequency, respectively. **e** the TOCs in ENZ region. The top panel of the diagram shows the values of thermo-optic effects of the O-band, C-band, 2-μm, and non-ENZ samples. The bottom panel of the diagram is the ENZ-induced thermo-optic effect enhancement with respect to the non-ENZ sample. The colored rectangles denote the full-width-at-half-maximum (FWHM) bandwidth (BW) of the corresponding curve. The colored vertical lines in **a** and **e** denote the corresponding ENZ frequencies.

Note 1.2. The detailed data on the thermal stability of samples and the reproducibility of the results are demonstrated in Supplementary Note 6.5.

## ENZ-induced enhancement of thermo-optic effects

The refractive index and its tunability are of great relevance to the design of ENZ-enabled nanophotonic devices, providing information on optical impedance, phase matching and reflections. A significant index change, as high as 170%, induced by ENZ conditions was demonstrated by optical excitation[7], and here we show that also the conventional thermo-optic effect is greatly enhanced—in the ENZ regime—by nearly an order of magnitude with ultrabroad bandwidth.

By utilizing the relation $\varepsilon_r = n^2 - k^2$ and $\varepsilon_i = 2nk$, with $k$ being the extinction coefficient, the full spectra of refractive indices at room temperature are depicted in Fig. 3a, with enlarged details for temperature comparison near the ENZ frequencies in Fig. 3b–d. As can be read from Fig. 3a, there exists a frequency band near the ENZ frequencies where the near-zero-index (NZI, $n < 1.0$) condition is satisfied. The NZI condition not only requires the $\varepsilon_r$ to be sufficiently small, but also $\varepsilon_i$ to be small enough in order not to make $n$ exceed 1.0. The lower $\nu_{ENZ}$ samples tend to have a higher $\varepsilon_i$, which results in a narrower NZI band. The refractive indices show a weak dependence on temperature, $n$ increasing with temperature. We also see that samples with higher $\nu_{ENZ}$ (higher $N$) have a lower $n$. The contrast between ITO and air being larger, higher bandwidth and stronger reflection are expected at normal incidence for such samples (see Supplementary Fig. 7).

While the temperature dependence of $\nu_{ENZ}$ and $n$ might seem small and trivial, this is not the case for the thermo-optic coefficient (TOC). We present in Fig. 3e the infrared spectrum of the TOCs of ENZ ITO (TOC$_{ENZ}$) and that of the non-ENZ sample (TOC$_{nonENZ}$). The TOC is calculated from the $n$ variation using TOC $= dn/dT$ for $\Delta T > 60\,°C$ margin. The absolute TOC value of non-ENZ ITO is found to be in the order of $10^{-5}\,K^{-1}$, which is consistent with the previously reported values[46]. To get a better appreciation of the TOC of ENZ samples and eliminate frequency-dependent non-ENZ-factor-induced variations, we define the TOC enhancement factor as $\mathcal{F}_{TOC}$:

$$\mathcal{F}(\nu, \nu_{ENZ})_{TOC} = \frac{TOC_{ENZ}}{TOC_{nonENZ}}. \tag{2}$$

The spectra of $\mathcal{F}_{TOC}$ are presented in the lower panel of Fig. 3e. We see that the behavior becomes more symmetrical across the ENZ region and that all three ENZ-related curves show a TOC close to an order of magnitude (660–955%) enhancement.

This can be understood by expressing the difference in $n$ between two temperatures as:

$$\Delta n = \frac{\sqrt{2}}{2}\left[\sqrt{\varepsilon_r' + \sqrt{\varepsilon_r'^2 + \varepsilon_i'^2}} - \sqrt{\varepsilon_r + \sqrt{\varepsilon_r^2 + \varepsilon_i^2}}\right]$$
$$\overset{\Delta\varepsilon_i \to 0}{\approx} \frac{\Delta\varepsilon_r}{\sqrt{\varepsilon_r + \Delta\varepsilon_r} + \sqrt{\varepsilon_r}} \approx \frac{\Delta\varepsilon_r}{2\sqrt{\varepsilon_r}}, \tag{3}$$

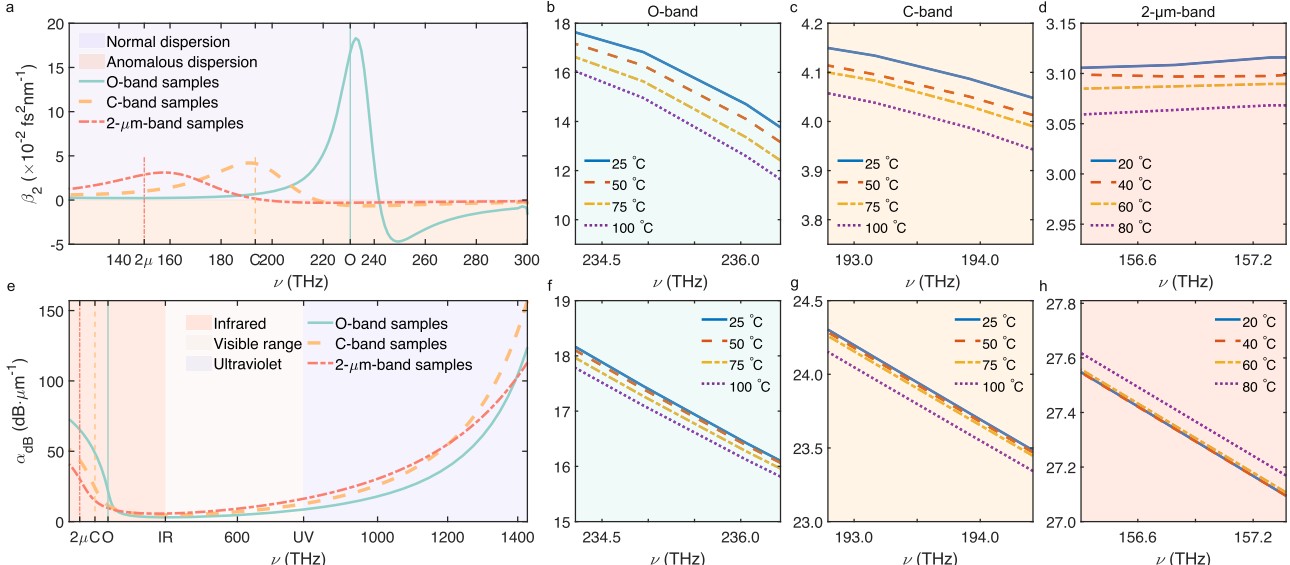

**Fig. 4 | Spectra of group velocity dispersion (GVD) and absorption. a** Full GVD spectrum of the O-band, C-band, and 2-μm-band samples. **b–d** The variations of GVD with temperature for these three sets of samples near the epsilon-near-zero (ENZ) frequency, respectively. **e** Full absorption spectrum of the three sets of samples. **f–h** The corresponding absorption variations with temperature near the ENZ frequency. In **a** and **e**, the colored vertical lines indicate the exact ENZ frequencies of the corresponding curves.

where $\varepsilon_r' = \varepsilon_r + \Delta\varepsilon_r$ and $\varepsilon_i' = \varepsilon_i + \Delta\varepsilon_i$ are the real and imaginary parts of the permittivity when temperature of the sample changes. The condition $\Delta\varepsilon_i \to 0$ is experimentally confirmed in the ENZ region (Supplementary Note 1.2). We can see that as $\varepsilon_r \to 0$ the conventional thermo-optic effect $dn/dT$ is enhanced. On the other hand, the bandwidth is related to the wide spectral regime of NZI. It is worth mentioning that the approximation made to obtain the last step of Eq. (3) serves to provide an intuitive idea of how TOC is enhanced by ENZ conditions. Only the first step of Eq. (3) is valid to cover the entire ENZ range.

Physically, the NZI and ENZ behaviors of ITO strongly rely on electron resonances, the plasma frequency $\omega_p$ described in Eq. (1). When heating, the electron resonances under high carrier concentration are more prone to be influenced by temperature in ENZ samples than the non-ENZ ones, yielding a more dramatic refractive index change, as is also experimentally confirmed from the difference between the three ENZ samples.

Note that the spectral locations of the $\mathcal{F}_{TOC}$ peak values are 236.06, 193.92, and 162.40 THz for O-band, C-band, and 2-μm-band samples, respectively, which exhibit a small offset with respect to their ENZ frequencies. This may be ascribed to the contribution of the loss variation term $\Delta\varepsilon_i$.

### Thermo-optic response of dispersion and loss

Besides the refractive index and TOC, other key optical parameters influencing many propagation dynamics are the GVD and the propagation loss $\alpha_{dB}$. The GVD is defined as[47]:

$$\beta_2(\omega) = \frac{1}{c} \cdot \frac{d^2(n \cdot \omega)}{d\omega^2}. \qquad (4)$$

In the field of ENZ, the GVD was previously numerically found and theoretically analyzed[48,49]. Based on our experimental data and Eq. (4), we derive $\beta_2$, which is plotted in Fig. 4a with its temperature-dependence in Fig. 4b–d. In Fig. 4a, the $\beta_2$ curves peak near $\nu_{ENZ}$, as a result of the rapid change in $\varepsilon_r$ and $n$. The peak values are 0.18, 0.04, and 0.03 fs² nm⁻¹ for O-band, C-band, and 2-μm-band samples, respectively, which agree well with the previous reports of ≈0.2 fs² nm⁻¹, and is around 4 orders of magnitude larger than in conventional optical materials like silica fibers[47]. All samples exhibit an

oscillatory profile traversing the normal dispersion ($\beta_2 > 0$) and the anomalous dispersion regimes ($\beta_2 < 0$), as the frequency increases. As also can be observed in Fig. 4b–d, the temperature modulation of $\beta_2$ is small and becomes weaker with lower $N$. This is attributed to a weaker electron resonance near the $\nu_{ENZ}$ for lower $N$ samples. The well-known Berreman mode and ENZ mode[50] cannot be supported due to the larger thickness and the non-flat dispersion induced by Drude resonance.

The loss assessment is another parameter of significance. The propagation loss per unit of length is expressed as[47]:

$$\alpha_{dB} = 10(\log_{10} e)\frac{\omega}{nc}\Im[\chi^{(1)}(\omega)] \approx 4.343\frac{\omega\varepsilon_i}{nc}, \qquad (5)$$

where $\Im$ denotes the imaginary part and $\chi^{(1)}$ is the first-order susceptibility. We plot in Fig. 4e $\alpha_{dB}$ derived from the experimental data. The high absorption of the ultraviolet regime is due to the Lorentz resonance, while the high absorption near $\nu_{ENZ}$ is from plasma resonance, with stronger absorption experienced by samples with higher $N$. The temperature variations, as shown in Fig. 4f–h, again have a very weak influence on the loss, as expected, noting that $\alpha_{dB} \propto (2\omega/c)k = (2\omega/c)(n^2 - \varepsilon_r)^{1/2}$ and that $n^2$ and $\varepsilon_r$ have trivial temperature modulation.

### Thermo-optic nonlinearity enhanced by ENZ conditions

As the thermal counterpart of the well-studied third-order Kerr nonlinearity[7,51], the thermo-optic nonlinearity can be dominant[47] at picosecond to continuous-wave (CW) operations[18], especially when the ENZ-enhanced TOC is involved. The thermo-optic nonlinearity has the form $n = n_0 + n_2^{(th)}I$ where the thermo-optic nonlinear-index coefficient $n_2^{(th)}$ reads[47]:

$$n_2^{(th)} = \left(\frac{dn}{dT}\right)\frac{\alpha r^2}{\kappa} = TOC \cdot \frac{\alpha r^2}{\kappa}, \qquad (6)$$

where $\alpha = \alpha_{dB}/(10\log_{10} e)$ is the loss in linear scale, $r$ is the beam radius, and $\kappa$ is the thermal conductivity of the ENZ material, which for ITO is 10.2 W m⁻¹ K⁻¹[52]. Resembling the classical Kerr nonlinearity, where in ENZ it is reported that the loss[53] and the near-zero permittivity[7] are the

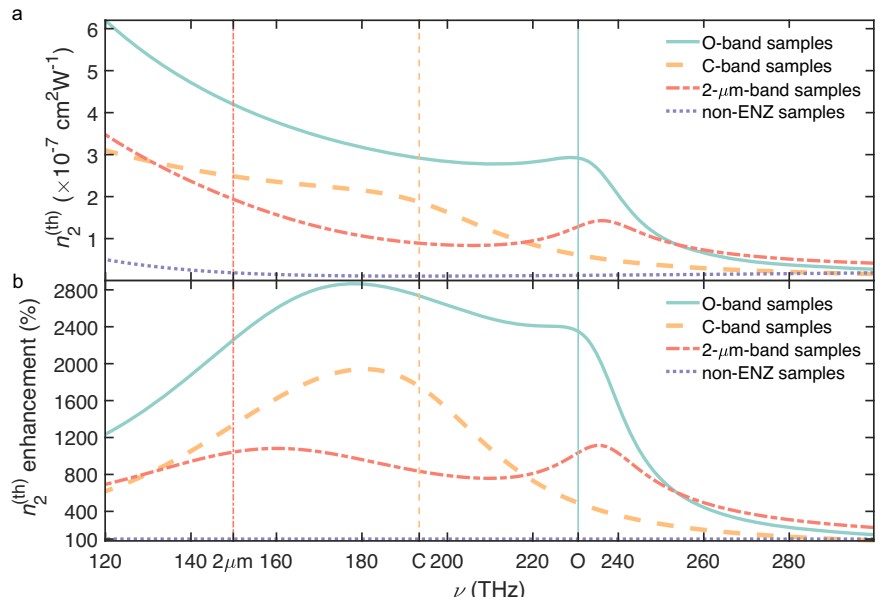

**Fig. 5 | Thermo-optic nonlinearity near epsilon-near-zero (ENZ) region. a** The thermo-optic nonlinear-index coefficients of the O-band, C-band, 2-μm-band ENZ and non-ENZ samples. **b** The ENZ-induced enhancement of the thermo-optic nonlinearity. The colored vertical lines indicate the exact ENZ frequencies of the corresponding curves.

major sources of the remarkably high Kerr nonlinear-index coefficient $n_2^{(el)}$, here in the case of the thermo-optic nonlinearity, the ENZ-enhanced TOC and loss are the main drivers for a large $n_2^{(th)}$.

We plot in Fig. 5a. the thermo-optic nonlinear-index coefficients near the ENZ for our three ENZ ITO samples together with the non-ENZ sample. The enhancement, expressed similarly as for the TOC enhancement is presented in Fig. 5b. The results are shown for a given chosen value of $r = 0.5\,\mu m$, comparable with the typical transverse mode size in integrated photonics. The ENZ samples show a broadband 1–2 orders of magnitude enhancement over the non-ENZ ITO. The corresponding values at the level of $10^{-7}\,cm^2\,W^{-1}$, which is nearly 7 orders of magnitude stronger than the low-loss silicon nitride platform[54] ($\approx 1.4 \times 10^{-14}\,cm^2\,W^{-1}$ under the same beam radius).

Previously, it is formulated theoretically that the commonly considered slow and often dismissed thermo-optic nonlinearity can be transitioned to an ultrafast one by plasmonic enhancement and surface plasmon polariton (SPP) with an optical excitation of tens of picoseconds, which is within the modern signal processing regime[55]. It is also theoretically analyzed that the electrons in the Drude materials, such as ITO, can heat up more and cool down faster compared with noble metals[56], which could benefit the potential ultrafast applications requiring the much stronger thermal-optic nonlinearities[55].

In this work, plasmonic enhancement exists in the ITO samples in an even stronger form, the ENZ condition enabled by high electron concentration and resonance. Using the TOC experimental data, we show that it is possible to obtain an even faster thermo-optic time scale within which the thermo-optic nonlinearity still dominates, without involving the nanoscale geometry and SPP condition satisfaction.

For integrated or nanophotonic ENZ devices, the exhibited optical dynamics under CW, longer pulse, and ultrashort pulse operations can be very different[11,18]. To estimate the operation boundary in terms of pulse width, a minimal pulse width $t_{min}$ can be defined[47], above which $n_2^{(th)}$ dominates, and below which $n_2^{(el)}$ is more prominent:

$$t_{min} \leq \frac{n_2^{(el)} C_{V,ITO}}{TOC \cdot \alpha}, \tag{7}$$

where $n_2^{(el)}$ values used for O-band[7], C-band[42], and 2-μm-band[42] are $2.58 \times 10^{-12}$, $2.07 \times 10^{-12}$, and $5.70 \times 10^{-13}\,cm^2\,W^{-1}$, respectively. $C_{V,ITO}$ is the volumetric heat capacity of ITO estimated to be $2.58\,MJ\,m^{-3}\,K^{-1}$[57]. $t_{min}$ yields a value in the range of 157–677 fs for these infrared-band ENZ ITOs, which are significantly faster than the typical $\approx 30$ ps estimation[47] of the conventional optical materials. It is worth noting that the ultrafast $t_{min}$ does not indicate the possibility of tuning and modulating temperature on the hundred-femtosecond scale, but a boundary at which the thermo-optic nonlinearity should not be neglected—the heat does not necessarily come only from the pulse, but can also be the environment. Therefore, from scientific and engineering points of view, any operation that falls inside this temporal range should consider the effect of $n_2^{(th)}$ alongside $n_2^{(el)}$.

## Results and discussion

Thermal management in optical communication systems can be a challenge for the data centers[56] involving integrated photonic devices. We have analyzed the thermo-optic ENZ behavior of sampled operating in the standard O-band, C-band, and 2-μm-bands. From the perspective of on-chip integration, the ENZ band and working temperature should be considered carefully. Due to a low dopant activation level, the 2-μm-band ENZ TCOs might be permanently damaged due to unrecoverable change in their internal structures even under the typical $T_{junction(max)}$. If applications are designed using the 2-μm-band, active heat dissipation countermeasures are needed. For narrowband NZI applications with low reflection (see Supplementary Note 4), the 2-μm-band ENZ TCOs is preferred, while for relatively broadband operation or when high reflection is needed, the O-band ones should work better. For nanodevice and nonlinear integration purposes, most of the optical parameters except the ENZ-enhanced TOC and $n_2^{(th)}$ at all bands are expected to be relatively temperature-insensitive but highly dispersive such that an accurate balance between the linear and nonlinear parameters should be carefully designed.

It was recently found that thermo-optic nonlinearities can be used to obtain analog electromagnetically induced transparency[58–60], which allows to achieve slow light and therefore to increase the light-matter interaction time, especially in integrated photonic platforms. In light of

our findings, ENZ materials could then represent in this context a key element for achieving on-chip group delays, thanks to their high thermo-optic nonlinearity, especially whereas broadband operation is required.

Being closely related to the PICs, the presented series of thermo-optic ENZ effects can also be useful in designing other ENZ-enabled integrated photonic devices and components in electro- and all-optical systems, such as thermo-optic modulators, switches, and attenuators, on-chip temperature sensors, interferometric devices, tunable optical filters, integrated phase shifters, beam steering components, and the fine control of on-chip signal processors[43].

In this work, we experimentally studied the total thermo-optic ENZ effects for the first time at the typical maximum junction temperature below annealing threshold. We report the characteristics of temperature-dependent linear and nonlinear-related optical properties considered for applications involving packaged highly integrated photonic systems, covering ENZ frequencies from the telecommunication O-band, C-band, and the 2-µm-band using ITO nanolayers.

Remarkably, an unprecedented high enhancement over the conventional thermo-optic effect is observed in the ENZ regions, with an ultrabroad FWHM bandwidth. The large oscillatory GVD is verified near the ENZ frequencies, matching previous theoretical predictions. We found for the first time that, resembling its Kerr nonlinearity counterpart, the thermo-optic nonlinearity shows an enormous enhancement in the ENZ zone over the conventional non-ENZ ITO. Due to the huge $n_2^{(\text{th})}$ and an ultrafast pulse-width threshold $t_{\min}$, the effect of the thermo-optic nonlinearity can be prominent for ENZ-enabled photonic PICs operating at sub-picosecond and femtosecond regime.

Our work reports novel physical phenomena and offers important parametric and practical references for applications of ENZ-based devices in PICs where heating is inevitable. Additionally, the revealed thermo-optic ENZ effects might provide a deeper understanding and insight into the field of ENZ science and materials as well as a new platform for slow-light on-chip applications and photonic emulation of other physical phenomena.

## Methods

### Sample preparation
All samples are fabricated using 99.99% purity 10 wt% ITO target by DC magnetron sputtering. The O-band and 2-µm-band samples are commercial and fabricated using a 160-kW continuous flow with 20% power, while the C-band and the non-ENZ normal samples are fabricated in a research cleanroom using 150 W power. After sputtering, the O-band samples are annealed in a vacuum for 30 min under 400 °C; the C-band samples are annealed in vacuum for 2 h under 350 °C; the 2-µm-band and the non-ENZ samples are not annealed. Although the conditions are the same except annealing within the two pairs, i.e., the O-band and C-band samples, as well as the 2-µm-band and non-ENZ samples, the outcomes are different due to the difference between factory and cleanroom equipment and the detailed operations used. This also indirectly reveals the well-acknowledged difficulty of fabricating an ENZ material with a specific ENZ frequency.

### Thermal control and engineering
For the thermal control, the 5.1 W Thorlabs TECD2S Peltier module, the AD590KF temperature sensor, and the 12 W/2.1 A Thorlabs TED200C temperature controller form a feedback loop. Similar solutions are used in PIC experiments with a precision of 0.01 ℃. The aluminum sample holder plate is computer-numerical-controlled (CNC)-machined and polished. The $T_{\text{sample}}$ versus $T_{\text{sensor}}$ verification is done by a thermocouple module of an Agilent U1272A at the center of a reference glass sample with the same thickness. The design is highly efficient with a maximum sensor-sample temperature difference of ≈3 °C (including a nominal ±1 °C error of the temperature sensor) at high temperatures. The cooling power and heat capacitance of the

shown heat sink can be considered irrelevant because when it is placed on the ellipsometer stage, the whole instrument acts as a heat sink. This thermal design could serve as a relatively universal reference engineering design due to its simplicity and low dependence on materials and structures. We also present a reference sample stage design for free-space optics experiments in Supplementary Note 2, whose efficiency is lower due to openings and the thermal conductivity of ITO.

### Ellipsometry
The ellipsometers used are J.A. Woollam RC2 (spectral range: 210–2500 nm) for the thermo-optic experiments and J.A. Woollam M2000-UI (spectral range: 246–1690 nm) for the post-fabrication characterization and verification. Between each measurement, the sample is given 2 min to stabilize at its set temperature, which is far sufficient for our thermal stage (see Supplementary Note 2). The measurements are performed by 5 °C a step up to 100 °C and cooled down rapidly from 80 °C to 20 °C. The temperature control, stabilization, and monitoring are achieved by the commercial temperature controller Thorlabs TED200C.

Besides $N$ (see Supplementary Fig. 4) that can be extracted by the Drude model in Eq. (1), ellipsometry can also provide estimations on the change in electrical properties like resistivity extraction[35] (see Supplementary Fig. 5a). From the Hall effect measurement, the trend of resistivity change (see Supplementary Fig. 5b) agrees with the ellipsometry estimation.

### Hall effect measurement
The electrical properties (resistivity and mobility) of the samples (see Supplementary Note 4) are acquired using a Nanometrics HL5500PC under a magnetic field of 0.501 T, a current of 5 mA, and an ambient temperature of 26.19 °C.

## Data availability
The experimental data used in this study are available in the Zenodo database under accession code https://doi.org/10.5281/zenodo.10148545.

## Code availability
The codes used to produce the plots of this paper are available at https://doi.org/10.5281/zenodo.10148545.

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

## Acknowledgements

This work is supported by the Swiss National Science Foundation (Grant No. 200021-188605) (C.-S.B.), the Basic and Applied Basic Research Foundation of Guangdong Province (Grant No. 2021A1515012176) (Q.L.), and Tsinghua Shenzhen International Graduate School-Shenzhen Pengrui Young Faculty Program of Shenzhen Pengrui Foundation (Grant No. SZPR2023008) (H.F.).

## Author contributions

J.W. conceived the original idea of this work. M.C. designed the thermal engineering solutions. J.W. conducted the main experiments. C.H. fabricated the experimental samples with preliminary testing. F.Y. performed the Hall effect measurements. L.L. provided the analyses on degenerate semiconductor theory. Q.L., H.Y.F., L.L., S.Z., and C.-S.B. provided the experimental resources. All authors took part in analyzing the data. J.W. wrote the paper with inputs from others. M.C., Q.L., and C.-S.B. provided in-depth reviews and discussions in revising the paper. Q.L. supervised the experiments conducted at Peking University. C.-S.B. supervised the project and experiments conducted at EPFL. All authors have proofread the paper.

## Competing interests

The authors declare no competing interests.
