## [Peer Review File · Nature Communications]

Thermo-optic epsilon-near-zero effectsREVIEWER COMMENTS

Reviewer #1 (Remarks to the Author):

The author examined the temperature dependent optical properties of ITO films by using ellipsometry. The results show that the ITO film exhibit highly temperature dependent linear and nonlinear optical response. This work is interesting and the results could be useful for practical applications. My comments are as follows.

1. Please provide the optical transmittance/reflectance curve for the different ITO films used un this present work.
2. What is the thickness of the ITO film. Please given a SEM image for the cross-section of the sample.
3. According to my experience, the electric performance of ITO films could be sensitive to the thermal treatment. Is it possible to provide experimental curves recorded at the same temperature after several heating-cooling cycles (below 100 degC)?
4. As the same target is used for the fabrication of the ITO films with different optical properties, their different carrier concentrations should be ascribed to many factors that are sensitive to annealing. Experimental details for the fabrication process as well as electric characterizations are important for better understanding of these materials.
5. It is shown that the peaks are located close to ENZ wavelengths for the measured samples in Fig. 3e. Please give the peak positions in Fig. 3e and comment on its relation with ENZ wavelength.
6. In this work, the thermo-optic nonlinearity is derived from eq. 6 based on the spectral curves. Is it possible to determiner thermo-optic nonlinearity by other method, considering that measurement of refractive index is based on curve fitting and is not so accurate?
7. Please list the Drude parameters for the three films examined at different temperatures in this work. This can be derived from the ellipsometry curves and also can be determined by direct electric measurement.

Reviewer #2 (Remarks to the Author):

The authors present a very thorough and detailed study of the thermo-optic effect in ITO with varying ENZ frequency. They demonstrate an impressive enhancement of the thermo-optic effect in the ERNZ region. The methodology is well chosen and all conclusions are supported by the data presented. I think this has the potential to be an important source for others working of the field and the authors provide very clear explanations of the physics underlying the thermo-optioc response in ITO. I think this is an excellent paper, well suited to this journal.

There are some minor concerns that I would ask the authors to address/consider.

1. The final step in eqn 3 assumes that $\Delta\epsilon$ is much smaller than ϵ . I question if this is still valid for the strong observed thermo-optic effect when occurring in the ENZ region. For reference, O. Reshef et al (Optics Letter 42, 3225) showed that a similar approximation often made in the description of the Kerr effect is no longer valid. I would also like to know if a similar issue would or could arise for the derivation of eqn.6

2.I might have missed it, but I could not find any reference to the thickness of the deposited ITO films. This should be stated and a brief discussion included if we are in the bulk or thin film (e.g. presence of Berreman mode) regime.

3. Did the ellipsometry analysis allow for grading of the permittivity within the ITO> Or the

presence of surface roughness? I think this should be briefly discussed (even if just to state that no grading was observed). I think this point does not need to be in the main paper, but would be well placed in the supplementary information.

Response Letter

Re: manuscript: NCOMMS-23-45855

Title: “Thermo-optic epsilon-near-zero effects”

by Jiaye Wu*, Marco Clementi, Chenxingyu Huang, Feng Ye, Hongyan Fu, Lei Lu, Shengdong Zhang, Qian Li*, and Camille-Sophie Brès*

We would like to express our gratitude to the Reviewers for their valuable comments and suggestions. We have addressed all the comments in the resubmitted manuscript. Our point-by-point response and a summary of the corresponding changes are following below. We first cite the comments and then respond to them. The changes are highlighted in the revised manuscript and supplementary information.

-----Reviewer Comments-----

Reviewer #1 (Remarks to the Author):

The author examined the temperature dependent optical properties of ITO films by using ellipsometry. The results show that the ITO film exhibit highly temperature dependent linear and nonlinear optical response. This work is interesting and the results could be useful for practical applications. My comments are as follows.

Reply: We thank the Reviewer for the recommendation and the recognition on the value of our work.

1. Please provide the optical transmittance/reflectance curve for the different ITO films used un this present work.

Reply: We thank the Reviewer for the nice suggestions. In the revised version of supplementary information, we added the new “**Supplementary Note 6 Sample information**” to discuss detailed information on our samples. In this added section, we dedicate a Subsection “**Supplementary Note 6.2 Thin film characteristics at room temperature: transmittance, reflectance, and absorbance**” displaying these linear optical properties of our samples (O-band, C-band, 2- μm -band ENZ ITO, and non-ENZ ITO) at normal incidence under room temperature.

In this Response Letter, we show the new Supplementary Figure 9 from Supplementary Note 6.2 as below:

Supplementary Figure 9 The full spectra of transmittance, reflectance, and absorbance of the O-band, C-band, 2- μ m-band ENZ, and non-ENZ ITO samples.

2. What is the thickness of the ITO film. Please given a SEM image for the cross-section of the sample.

Reply: We thank the Reviewer for the question. The nominal thickness of the ITO film is 130 nm for all samples. We added the thickness in the main text, “We prepare 4 different types of ITO samples with a nominal thickness of 130 nm (see *Supplementary Note 6* for sample characterizations) ...” when we first describe our samples.

The actual thickness might vary with fabrication machines, techniques, precision, and annealing. We measured the thickness of the ITO film using the ellipsometer and we added the relevant information in the new **Supplementary Note 6.1** *Fabrication details*, shown below.

Supplementary Note 6.1 **Fabrication details**

Fabrication method: 99.99% purity 10 wt% ITO target by DC magnetron sputtering.

Nominal thickness: 130 nm (actual thickness might vary with fabrication machines, techniques, precision, and annealing).

ITO samples ¹	Actual thickness ²	Fabrication environment	Sputtering power	Annealing conditions
O-band	138.06 \pm 0.236 nm	continuous flow ³	160 kW, 20% power	400 °C, 30 minutes in vacuum
C-band	162.91 \pm 0.666 nm	research cleanroom	150 W	350 °C, 2 hours in vacuum
2- μ m-band	127.63 \pm 0.178 nm	continuous flow	160 kW, 20% power	no annealing
non-ENZ	146.49 \pm 0.116 nm	research cleanroom	150 W	no annealing

Note:

1. The electrical properties and their temperature dependence can be referred to in Supplementary Note 4
2. Thickness obtained from ellipsometry. The deviations in thickness do not affect the results presented in this work, since both the TOC and the thermo-optic nonlinearity are thickness-independent.
3. This technique is commercial and allows for large-area deposition and large-scale high-stability production.

Additionally, following the suggestion by the Reviewer, we have retrieved and added the SEM image of the cross-section of the samples in the new **Supplementary Note 6.4** *Cross-sectional scanning electron microscope (SEM) images*, shown below:

Supplementary Note 6.4 Cross-sectional scanning electron microscope (SEM) images

By the state-of-the-art SEM imaging technology, we took the cross-sectional photographs of our samples shown in Supplementary Fig. 10 as a verification of the ellipsometry results. The thickness values from the two instruments are very close (*c.f.*, Supplementary Note 6.1), indicating a sufficiently good fit of the ellipsometer model. The images are filmed by a HITACHI SU8010 cold-field emission SEM with $\times 100k$ magnification.

Supplementary Figure 10. The cross-sectional SEM images of the ITO samples. **a** O-band, **b** C-band, **c** 2- μ m-band ENZ, and **d** non-ENZ.

3. According to my experience, the electric performance of ITO films could be sensitive to the thermal treatment. Is it possible to provide experimental curves recorded at the same temperature after several heating-cooling cycles (below 100 degC)?

Reply: We thank the Reviewer for the suggestion. To test the thermal stability and degradation of the ENZ ITO samples, and to demonstrate the reproducibility of the presented experiments, we perform two extra cycles of the heat-up and cool-down processes in addition to our original data.

The results are illustrated in Supplementary Figure 11 (shown below) in the new **Supplementary Note 6.5** *Stability over additional heat-up and cool-down cycles*, with the O-band and C-band samples exhibiting

recoverability, while the 2- μm -band samples show irreversible degradation over cycles due to the occurrence of annealing. Note that this behavior is consistent with our previous observations.

We also added in the main text, at the end of the *Temperature-dependent ENZ frequencies* Section, “The detailed data on the thermal stability of samples and the reproducibility of the results are demonstrated in Supplementary Note 6.5.”

Supplementary Figure 11 The change of ENZ frequencies over additional heat-up and cool-down cycles of the O-band, C-band, and 2- μm -band ENZ ITO samples.

4. As the same target is used for the fabrication of the ITO films with different optical properties, their different carrier concentrations should be ascribed to many factors that are sensitive to annealing. Experimental details for the fabrication process as well as electric characterizations are important for better understanding of these materials.

Reply: We thank the Reviewer for the comments. Following the suggestion by the Reviewer, we have added the new **Supplementary Note 6.1** *Fabrication details* showing the processes such as deposition power, annealing, etc. The added information is shown below:

Supplementary Note 6.1 Fabrication details

Fabrication method: 99.99% purity 10 wt% ITO target by DC magnetron sputtering.

Nominal thickness: 130 nm (actual thickness might vary with fabrication machines, techniques, precision, and annealing).

ITO samples ¹	Actual thickness ²	Fabrication environment	Sputtering power	Annealing conditions
O-band	138.06±0.236 nm	continuous flow ³	160 kW, 20% power	400 °C, 30 minutes in vacuum
C-band	162.91±0.666 nm	research cleanroom	150 W	350 °C, 2 hours in vacuum
2- μ m-band	127.63±0.178 nm	continuous flow	160 kW, 20% power	no annealing
non-ENZ	146.49±0.116 nm	research cleanroom	150 W	no annealing

Note:

1. The electrical properties and their temperature dependence can be referred to in Supplementary Note 4
2. Thickness obtained from ellipsometry. The deviations in thickness do not affect the results presented in this work, since both the TOC and the thermo-optic nonlinearity are thickness-independent.
3. This technique is commercial and allows for large-area deposition and large-scale high-stability production.

The related electrical properties and their temperature dependence are already presented in **Supplementary Note 4** *Temperature dependence of carrier concentration, resistivity, mobility, and reflection*.

5. It is shown that the peaks are located close to ENZ wavelengths for the measured samples in Fig. 3e. Please give the peak positions in Fig. 3e and comment on its relation with ENZ wavelength.

Reply: We thank the Reviewer for the question. The TOC peaks for O-band, C-band, and 2- μ m-band samples are located at 236.06 THz, 193.92 THz, and 162.40 THz, respectively, which are all slightly lower than their respective ENZ frequencies of 235.64 THz, 193.83 THz, and 157.03 THz. This small offset is due to the contribution of the small value of loss change $\Delta\epsilon_i$ in the first line of Eq. (3).

In the revised version of our manuscript, we added the following discussion on the peak spectral locations after showing Eq. 3 at the end of the Subsection:

“Note that the spectral locations of the \mathcal{F}_{TOC} peak values are 236.06 THz, 193.92 THz, and 162.40 THz for O-band, C-band, and 2- μ m-band samples, respectively, which exhibit a small offset with respect to their ENZ frequencies. This may be ascribed to the contribution of the loss variation term $\Delta\epsilon_i$.”

6. In this work, the thermo-optic nonlinearity is derived from eq. 6 based on the spectral curves. Is it possible to determiner thermo-optic nonlinearity by other method, considering that measurement of refractive index is based on curve fitting and is not so accurate?

Reply: We thank the Reviewer for the question.

Indeed, there are other methods to obtain the thermo-optic nonlinearity, namely, by z-scan, which is also used to measure the $\chi^{(3)}$ -induced Kerr nonlinearity [1,2]. However, we discovered that the transmission-type of thermal sample stage generally has a lower efficiency than the reflection-type we used (*c.f.*, Supplementary Note 2). Moreover, the z-scan method, as a matter of fact, is also a curve-fitting method that does not yield direct results.

On the other hand, our method of obtaining the thermo-optic nonlinearity is in fact sufficiently accurate. The origin of the data, ellipsometry, as a well-acknowledged record of accuracy in acquiring thin film thickness, permittivity, and refractive index [3]. In Eq. (6), the thermo-optic nonlinearity is related to TOC, loss, and mode area. The TOC and the loss are results calculated from ellipsometry data, and they are bound by the causality connections of the Kramers-Kronig relations.

Therefore, we consider our presented method sufficiently adequate to support our findings.

References:

[1] M. Falconieri, *Thermo-optical effects in Z -scan measurements using high-repetition-rate lasers*, J. Opt. A: Pure Appl. Opt. **1**: 662, 1999.
 [2] R. Karimzadeh and N. Mansour, *Thermo-optic nonlinear response of silver nanoparticle colloids under a low power laser irradiation at 532 nm*, Phys. Stat. Sol. B, **247**: 365-370, 2010.
 [3] D.E. Aspnes, *The Accurate Determination of Optical Properties by Ellipsometry* in *Handbook of Optical Constants of Solids*, Chapter 5, Volume I, Pages 89-112, Academic Press, 1997.

7. Please list the Drude parameters for the three films examined at different temperatures in this work. This can be derived from the ellipsometry curves and also can be determined by direct electric measurement.

Reply: We thank the Reviewer for the suggestion. We have added the new **Supplementary Note 6.3 Drude parameters extraction** to show the extracted values:

Supplementary Note 6.3 Drude parameters extraction

By the Drude model² described in Eq. 1, and assuming the effective mass of the electron⁹ is $m^* = 0.38m_0$, the following Drude parameters can be extracted by using both the ellipsometry and the Hall effect data.

ITO samples	ϵ_b (dimensionless)	N (cm^{-3})	ω_{ENZ} ($\text{rad}\cdot\text{s}^{-1}$)	ω_p ($\text{rad}\cdot\text{s}^{-1}$)	γ ($\text{rad}\cdot\text{s}^{-1}$)
O-band	3.8085	1.00×10^{21}	1.48×10^{15}	2.90×10^{15}	1.12×10^{14}
C-band	3.7837	6.82×10^{20}	1.22×10^{15}	2.39×10^{15}	1.65×10^{14}
2- μm -band	3.3599	4.50×10^{20}	9.87×10^{14}	1.94×10^{15}	3.86×10^{14}

Reviewer #2 (Remarks to the Author):

The authors present a very thorough and detailed study of the thermo-optic effect in ITO with varying ENZ frequency. They demonstrate an impressive enhancement of the thermo-optic effect in the ERNZ region. The methodology is well chosen and all conclusions are supported by the data presented. I think this has the potential to be an important source for others working of the field and the authors provide very clear explanations of the physics underlying the thermo-optic response in ITO. I think this is an excellent paper, well suited to this journal.

Reply: We thank the Reviewer for the recommendation and the recognition on the value of our work.

There are some minor concerns that I would ask the authors to address/consider.

1. The final step in eqn 3 assumes that $\Delta\epsilon$ is much smaller than ϵ . I question if this is still valid for the strong observed thermo-optic effect when occurring in the ENZ region. For reference, O. Reshef et al

(Optics Letter 42, 3225) showed that a similar approximation often made in the description of the Kerr effect is no longer valid. I would also like to know if a similar issue would or could arise for the derivation of eqn.6

Reply: We thank the Reviewer for the question.

To clarify, we first introduce the assumptions made in Eq. (3). In Eq. (3), we assume $\Delta\varepsilon_i \rightarrow 0$ to obtain the approximation $\Delta n = \Delta\varepsilon_r/2\sqrt{\varepsilon_r}$, in order to more intuitively show the readers why ENZ conditions can enhance TOC. In fact, this quantity does influence the spectral locations of the peaks of TOC enhancement factors: the peaks exhibit a small offset with respect to the ENZ frequency where the ε_r is the smallest. We added a discussion on this at the end of the subsection:

“Note that the spectral locations of the \mathcal{F}_{TOC} peak values are 236.06 THz, 193.92 THz, and 162.40 THz for O-band, C-band, and 2- μm -band samples, respectively, which exhibits a small offset with respect to their ENZ frequencies. This may be ascribed to the contribution of the loss variation term $\Delta\varepsilon_i$.”

The last step of Eq. (3) is also used to only give the readers an intuitive idea of how TOC is enhanced. This approximation, however is not strictly valid when we have $\varepsilon_r \approx 0$, and under this condition one should refer to the first step of Eq. (3) for the exact formula. To clarify this, we added a brief description of Eq. (3) in the corresponding paragraph: “It is worth mentioning that the approximation made to obtain the last step of Eq. (3) serves to provide an intuitive idea of how TOC is enhanced by ENZ conditions. Only the first step of Eq. (3) is valid to cover the entire ENZ range.”

For the second question, Eq. (6) is not derived from Eq. (3), and it is not affected by the assumptions. Unlike [O. Reshef et al *Opt. Lett.* **42**, 3225, 2017], the Δn in thermo-optic nonlinearity has a different source. In that reference, the reason for n_2 to be no longer valid is that the n expansion diverges with small index. Here the thermo-optic nonlinearity is based on the thermo-optic effect, in which $n = n_0 + (dn/dT)T$ is not affected by the small index and exists in every optical material when temperature changes. The thermo-optic n_2 here is just an analogue to the electric n_2 , which relates Δn to light intensity instead of temperature.

We added the suggested reference as Ref. [51].

2.I might have missed it, but I could not find any reference to the thickness of the deposited ITO films. This should be stated and a brief discussion included if we are in the bulk or thin film (e.g. presence of Berreman mode) regime.

Reply: We thank the Reviewer for the question. The nominal thickness of the ITO film is 130 nm for all samples. We added the thickness in the main text, “We prepare 4 different types of ITO samples with a nominal thickness of 130 nm (see Supplementary Note 6 for sample characterizations).” when we first describe our samples.

The actual thickness might vary with fabrication machines, techniques, precision, and annealing. We measured the thickness of the ITO film using the ellipsometer and we added the relevant information in the new **Supplementary Note 6.1 Fabrication details**, shown below. The ellipsometer estimate has also been verified with cross-sectional SEM imaging, yielding consistent results, which are now shown in **Supplementary Note 6.4**.

Supplementary Note 6.1 Fabrication details

Fabrication method: 99.99% purity 10 wt% ITO target by DC magnetron sputtering.

Nominal thickness: 130 nm (actual thickness might vary with fabrication machines, techniques, precision, and annealing).

ITO samples ¹	Actual thickness ²	Fabrication environment	Sputtering power	Annealing conditions
O-band	138.06±0.236 nm	continuous flow ³	160 kW, 20% power	400 °C, 30 minutes in vacuum
C-band	162.91±0.666 nm	research cleanroom	150 W	350 °C, 2 hours in vacuum
2- μ m-band	127.63±0.178 nm	continuous flow	160 kW, 20% power	no annealing
non-ENZ	146.49±0.116 nm	research cleanroom	150 W	no annealing

Note:

1. The electrical properties and their temperature dependence can be referred to in Supplementary Note 4
2. Thickness obtained from ellipsometry. The deviations in thickness do not affect the results presented in this work, since both the TOC and the thermo-optic nonlinearity are thickness-independent.
3. This technique is commercial and allows for large-area deposition and large-scale high-stability production.

The Berreman mode requires the sample to be a few nanometers thick (Ref. [50]) such that $\omega \approx \omega_L$ and a flat dispersion. Due to our rapidly changing dispersion curve of this thicker media where Drude model holds, the Berreman mode / ENZ mode is unlikely to be supported. We have added a small discussion after the description of the dispersion, “The well-known Berreman mode and ENZ mode [50] cannot be supported due to the larger thickness and the non-flat dispersion induced by Drude resonance.”

[50] S. Vassant et al., *Berreman mode and epsilon near zero mode*, Opt. Express **20**(21): 23971, 2012.

3. Did the ellipsometry analysis allow for grading of the permittivity within the ITO? Or the presence of surface roughness? I think this should be briefly discussed (even if just to state that no grading was observed). I think this point does not need to be in the main paper, but would be well placed in the supplementary information.

Reply: We thank the Reviewer for the suggestions. Yes, the ellipsometry analysis allow for the graded index (% inhomogeneity) and the surface roughness, and the factors has already been accounted for in our work.

The inhomogeneity and surface roughness analyses allow for comparison among different fabrication techniques and qualities, which might be useful for the reference of future experiments. We added the new **Supplementary Note 6.6 Uniformity assessment of samples**, showing these non-uniformity values:

Supplementary Note 6.6 Uniformity assessment of samples

The fitting models of ellipsometry allow for the assessment of sample uniformity in terms of surface roughness and inhomogeneity. The non-uniformity of the samples has been taken into account in the experiments and it also provides additional information for the comparison among different fabrication methods.

ITO samples	Surface roughness	% Inhomogeneity (graded index model)	Sample type
O-band	6.16±0.131 nm	0.10	commercial
C-band	3.23±0.384 nm	11.60	research cleanroom
2- μ m-band	8.12±0.116 nm	-2.05	commercial

We have highlighted the changes in the revised version of our manuscript.

Thank you for your consideration,
Authors

REVIEWERS' COMMENTS

Reviewer #1 (Remarks to the Author):

Just one comment:

Please provide the experimental details for determining the transmission/reflection/absorption curves given in Supplementary Figure 9.

Reviewer #2 (Remarks to the Author):

The authors have addressed all my comments to my satisfaction.

Therefore I stand by my initial assessment that this is an excellent paper and was a pleasure to read.

Response Letter

Re: manuscript: NCOMMS-23-45855A

Title: “*Thermo-optic epsilon-near-zero effects*”

by Jiaye Wu*, Marco Clementi, Chenxingyu Huang, Feng Ye, Hongyan Fu, Lei Lu, Shengdong Zhang, Qian Li*, and Camille-Sophie Brès*

We would like to express our gratitude to the Reviewers for their valuable comments and suggestions. We have addressed all the comments in the resubmitted manuscript. Our point-by-point response and a summary of the corresponding changes are following below. We first cite the comments and then respond to them. The changes are made in the supplementary information.

-----Reviewer Comments-----

Reviewer #1 (Remarks to the Author):

Just one comment:

Please provide the experimental details for determining the transmission/reflection/absorption curves given in Supplementary Figure 9.

Reply: We thank the Reviewer for the recommendation. The experimental details for Supplementary Figure 9 is now provided in Supplementary Note 6.2 “The curves are calculated from the ellipsometry data and they agree with the theoretical predictions in our previous work⁹.”

Reviewer #2 (Remarks to the Author):

The authors have addressed all my comments to my satisfaction.

Therefore I stand by my initial assessment that this is an excellent paper and was a pleasure to read.

Reply: We thank the Reviewer for the recommendation and the appreciation of our work.

Thank you for your consideration,
Authors